# Impact of Early Weaning on Development of the Swine Gut Microbiome

**DOI:** 10.3390/microorganisms11071753

**Published:** 2023-07-05

**Authors:** Benoit St-Pierre, Jorge Yair Perez Palencia, Ryan S. Samuel

**Affiliations:** Department of Animal Science, South Dakota State University, Animal Science Complex, Box 2170, Brookings, SD 57007, USA; jorge.perezpalencia@sdstate.edu (J.Y.P.P.); ryan.samuel@sdstate.edu (R.S.S.)

**Keywords:** swine, weaning, microbiome, gut development

## Abstract

Considering that pigs are naturally weaned between 12 and 18 weeks of age, the common practice in the modern swine industry of weaning as early as between two and four weeks of age increases challenges during this transition period. Indeed, young pigs with an immature gut are suddenly separated from the sow, switched from milk to a diet consisting of only solid ingredients, and subjected to a new social hierarchy from mixing multiple litters. From the perspective of host gut development, weaning under these conditions causes a regression in histological structure as well as in digestive and barrier functions. While the gut is the main center of immunity in mature animals, the underdeveloped gut of early weaned pigs has yet to contribute to this function until seven weeks of age. The gut microbiota or microbiome, an essential contributor to the health and nutrition of their animal host, undergoes dramatic alterations during this transition, and this descriptive review aims to present a microbial ecology-based perspective on these events. Indeed, as gut microbial communities are dependent on cross-feeding relationships, the change in substrate availability triggers a cascade of succession events until a stable composition is reached. During this process, the gut microbiota is unstable and prone to dysbiosis, which can devolve into a diseased state. One potential strategy to accelerate maturation of the gut microbiome would be to identify microbial species that are critical to mature swine gut microbiomes, and develop strategies to facilitate their establishment in early post-weaning microbial communities.

## 1. Weaning Is a Critical Phase in Commercial Swine Production

Swine production is a major component of the livestock and agriculture sectors worldwide; in the US alone, the swine industry contributed over USD 57 billion to the American economy in 2021 [1]. For commercial swine operations, the production cycle can be divided into four main phases: nursing, nursery, growing, and finishing. Weaning, which is the transition from nursing to the nursery phase, is more traumatic in commercial swine production compared to other livestock systems because of the number of changes that are implemented without any gradual transitions. Indeed, young pigs with an immature gut are suddenly separated from the sow, switched from milk to a diet consisting of only solid ingredients, and subjected to a new social hierarchy from mixing multiple litters [2,3]. While weaning is also a stressful experience for other mammalian livestock species as a result of separation from the dam and changes in social structure, it typically occurs after a period during which young animals have had the opportunity to gradually adapt to solid feed. Thus, the newly weaned pig presents very unique challenges for swine producers, and transitioning piglets from the farrowing room to the nursery or wean-to-finish barn is an important foundational step toward producing full-market value finisher pigs [4].

Intestinal health has a significant impact on overall pig health, nutrient utilization, and ultimately growth performance. Aside from feed digestion and nutrient absorption, the gut host tissue is responsible for a variety of physiological and biochemical functions related to energy production, immune response, and barrier protection against antigens and pathogens [5]. Thus, any detrimental effects on pig gut health can impair their ability to use dietary nutrients, compromise their health status, and consequently reduce their production efficiency. Considering that pigs are naturally weaned between 12 and 18 weeks, the common practice in the modern swine industry of early weaning between two and four weeks of age increases challenges during this transition period. Indeed, young pigs have immature digestive and immune systems, which, in concert with the interruption in the supply of immunoglobulins and other components provided by milk, result in an increased susceptibility to pathogens, potentially leading to intestinal disorders and diseases [6]. Weaning is commonly accompanied by reduced feed intake, an increased incidence of diarrhea, poor growth performance, a greater susceptibility to diseases, as well as increased mortality (>10%), which ultimately cause economic losses for swine producers [7,8,9]. These negative effects not only impact the first week after weaning, as growing evidence supports that such early life stressors can alter the developmental trajectory of the architecture and functions of the gut, thus having long-term negative effects on pig productivity [10,11].

One of the main factors responsible for falling performance in the first weeks after weaning is low feed intake [12]. At weaning, pigs have no other choice than to adapt to a less digestible and palatable solid diet, resulting in a significant proportion of pigs that either do not consume any feed or have poor feed intake during the first days after weaning. Metabolizable energy intake can be reduced by 40% during the period immediately after weaning [13], and it may take approximately two weeks to achieve complete recovery to the pre-weaning energy intake. The mechanisms responsible for the reduced performance by low feed intake involve adverse morphological and functional changes in the intestine, such as the shortening of villi, hyperplasia of crypt cells, as well as damage to gut mucosal integrity [14,15,16]. As a result of these changes, there is also a decrease in brush-border enzyme activity, as well as a reduction in absorptive capacity, which are both also detrimental to gut function [17]. Furthermore, the increase in permeability makes the gut susceptible to the action of luminal antigens, resulting in inflammation and the potential risk of disease [15]. Thus, encouraging feed and water intake is critical to adapting weaned pigs to the new diet regiment. While piglets nurse frequently (~16–20 meals per day), weaned pigs eat infrequently, often visiting the feeder only once per day even when they have free access to feed [18]. Providing small amounts of feed on mats at least two times per day, as well as adjusting feeders and form, can help pigs in their transition to solid feed [19]. Preparing diets with fresh ingredients of high quality that provide optimal nutrient content is also beneficial. For instance, highly digestible feedstuffs, such as hydrolyzed or fermented soy protein [20,21,22], can be included in specialized diets for weaned pigs [23]. While they add to production costs, specialized ingredients have been shown to more effectively transition piglets from milk to dry, grain-based diets [24,25].

Weaning age is critical to promoting the successful development of a young pig, as weaning older animals can help prevent challenges from an immature gut [26]. For instance, the production of digestive enzymes, such as protease, amylase, maltase, and sucrase, steadily increases after approximately 4 weeks of age, whereas lactase production sharply declines [27,28]. These adaptations in enzyme production greatly help pigs in transitioning from milk to grain-based diets [27,29]. In addition, antibody protection gradually wanes from the moment a piglet receives immunoglobulins from colostrum, which is the first milk provided by the sow within a period of 12–48 h after the onset of parturition. In addition to energy and nutrients, colostrum also provides the developing neonate with immune protection during nursing [30,31]. As this initial antibody protection lasts until approximately three weeks of age [32], weaning piglets at this age can be detrimental, as the young animals are still developing their own adaptive immune response system [33].

## 2. Impact of Early Weaning on the Swine Gut

### 2.1. Intestinal Histology and Digestive Enzyme Activity

The integrity of the intestinal tissue is a key factor for the efficient digestion and absorption of nutrients. When evaluating intestinal morphology, primary features include villus height, crypt depth, and the ratio of villus height and crypt depth; a reduction in villus height and/or villus-to-crypt ratio indicates that the intestinal structure is impaired, and that intestinal digestion and absorption capacity have been compromised [34]. Pigs weaned at 21 or 28 days have been reported to exhibit a reduction in villus height by as much as 35%, as well as decreased crypt cell production [35,36]. Similarly, pigs weaned at 21 days of age showed a reduction in villus height and in the villus:crypt ratio on day 3 and day 7 post weaning, indicating a deterioration in intestinal morphology associated with weaning [37]; notably, villus height and the villus:crypt ratio did not return to preweaning status until day 14 post weaning [37]. Another report has shown that pigs weaned at 21 days of age had reduced villus height in the proximal jejunum at days 2, 5, and 8 after weaning, and, while improvements were observed, villus heights were still 23% lower at day 15 post weaning when compared to preweaning status [38].

During the suckling period, piglets exhibit a rapid development in digestive functions, especially during the first 24 h after birth [39]. Adequate nutrition provided by colostrum and milk promotes an increase in digestive enzyme activities such as lactase, protease, and lipase. However, after weaning, pigs experience a dramatic change in brush-border digestive enzyme activities that is mainly due to low feed intake and the change in diet composition [12,40]. While only a transient effect has been reported on maltase activity in pigs weaned at 21 days, intestinal lactase and amino-peptidase activities have been found to be dramatically lower, with reductions as low as 84%, until at least 15 days post weaning [7,38]. In addition, the expression of other enzymes essential to small intestinal functions, such as alkaline phosphatase, was also reported to be lower after weaning [41,42]. Together, these changes in enzyme activities and the alterations in intestinal histology negatively impact the digestive and absorptive capacity of the small intestine in weaned pigs.

### 2.2. Intestinal Barrier Function

As part of the first line of defense, intestinal barrier function is critical for protecting pigs from toxins, pathogens, or antigens that may be present in the intestinal lumen [43]. The intestinal barrier has four main components. The first consists of the intestinal epithelial cells and their physical association through tight junctions, which form a lining that blocks the paracellular passage of harmful microorganisms or substances from the intestinal lumen into the body. The second component consists of the intestinal mucus layer, a chemical barrier made of mucins and antimicrobial proteins that prevents the adhesion and proliferation of pathogenic microorganisms. The third barrier is composed of cells from the immune system, whose function is to recognize and eliminate pathogens. Finally, the microbial barrier, which consists of commensal and beneficial microorganisms as well as their metabolites, maintains conditions that minimize the risk of pathogenic colonization and proliferation [34,43,44].

Because the first two to three months are crucial for the development and maturation of the intestinal barrier in pigs, any disruptions during this period can have a long-term impact on gut health and function [44]. Accordingly, early weaning has been associated with impaired physical barrier function and increased intestinal permeability, which may result in the leaking of harmful microorganisms and compounds into internal tissue layers; this in turn can trigger inflammation and potentially cause systemic diseases [28,37,45]. In pigs weaned at 21 days of age, intestinal barrier function was reported to be defective at different timepoints within the first 14 days after weaning compared to preweaning [37,46,47]. In addition, the expression of mRNAs encoding for the gap junction proteins occludin, claudin-1, and zonula occludens-1 was reduced within the first few weeks after weaning [37]. Based on these and other studies, increasing the weaning age can help mitigate some of the detrimental effects of weaning stressors on intestinal barrier function. As continued low feed intake can predispose pigs to intestinal barrier dysfunction [15], promoting adequate feed intake after weaning can help prevent the loss of intestinal barrier function [12].

### 2.3. Intestinal Immunity

The intestine is the largest immunological organ, with up to 70% of immune cells located in the mucosal and submucosal areas [48]. The functions of immune cells such as intraepithelial lymphocytes, lamina propria lymphocytes, neutrophils, and macrophages are coordinated through expression of cytokines to produce a variety of effectors such as antibacterial peptides and immunoglobulins. While effective activation of the defense mechanisms is critical, an important component of immune function regulation is also to prevent hypersensitivity to antigens [34,49].

As complete development of the intestinal immune system occurs at seven weeks of age, standard weaning practices in the swine industry then transition pigs at an age when their immune system has not yet matured [50]. Indeed, early weaning has been associated with an upregulation in the expression of inflammatory cytokines in the gut, including interleukin-1b (IL-1b), interleukin-6 (IL-6), and tumor necrosis factor-α (TNF-α) within the first week post weaning [37,51]. Upregulation of pro-inflammatory cytokines can affect intestinal integrity and epithelial functions, resulting in increased intestinal permeability [52,53]. In addition to alterations in cytokine production, weaning has also been associated with an increase in the number of intestinal inflammatory T cells and mast cells [9,14,54]. Since early weaning can cause long-term alterations in pig intestinal immune responses, controlling and/or reducing intestinal inflammation may alleviate subsequent intestinal disorders induced by weaning stress [55].

## 3. Gut Microbiomes and Their Contributions to Their Host

Through their association with microorganisms, animals form ‘metaorganisms’, i.e., they not only consist of host cells that originated from a fertilized egg, but also include a wide variety of microbial species that are each adapted to thrive in different micro-habitats that form on various areas of the host’s body, such as in the gut, lungs, or skin [56]. Microbial species adapted to the same habitat assemble into communities, which allows them to benefit from the complementary metabolic capabilities of co-existing microorganisms in that environment [57]. Such microbial communities are commonly referred to as microbiota or microbiomes, and they are critically important for maintaining the health and well-being of their host [58,59].

Amongst the various microbial communities living in close association with animals, the gut harbors the most complex of microbiomes, as they include the highest species diversity and cell densities [60,61]. Gut microbiomes contribute to both the nutrition and health of their host. Indeed, intestinal microbial communities provide nutrients in the form of short-chain fatty acids (SCFAs, mostly acetate, propionate, and butyrate) as well as vitamins (such as vitamins B12 and K) from metabolizing plant structural polysaccharides and other compounds that cannot be broken down efficiently by enzymes encoded in animal genomes [62,63,64,65]. In addition to their roles in nutrition, gut microbiomes also contribute to disease resistance and health status of the host by competing with pathogens and by participating in the development of the host’s immune system [59,66,67]. Accordingly, associations between the pig gut microbiome and animal health or performance have been reported [68,69,70,71,72,73]. Other published studies have shown that manipulating the gut microbiome can improve feed efficiency and average daily gain [74,75,76,77], as well as improve herd health [78,79,80,81,82]. Insights on the impact of the gut microbiome on more specific traits, such as adiposity [83,84,85] or digestibility [86], have also been reported.

Thus, in light of our current understanding of gut microbiomes, optimizing their function represents an attractive strategy to improve performance efficiency and animal health. Indeed, past management practices, such as prophylactic use of antibiotics, have shown that targeting gut microbiomes can have positive impacts on animal production [87], and have provided support to further pursue development of alternative strategies such as prebiotics, probiotics, and postbiotics. Microbiome research has been revolutionized by the rapid development of different ‘omics’-type tools and platforms, as they have allowed more comprehensive investigations of complex microbial communities [88,89]. While great insights have been achieved so far, further investigations are still required to gain a deeper understanding of microbiome capabilities towards developing strategies to modulate their function for optimizing livestock production and animal health [63].

The early establishment and continued maintenance of an optimal gut microbiome have been shown to be an important determinant of an animal’s future health status and productivity [90]. In light of the increased recognition that early events taking place during microbiome development can affect its function at later stages [63,67], gaining a better understanding of the mechanisms that dictate or affect the development of gut microbiomes in young animals has been of very high interest. This is particularly true in food animal production systems, where ensuring the health of young animals while conforming to stricter restrictions on traditional management practices such as antibiotics use remains an ongoing challenge [55]. Within the swine industry, the standard practice of abruptly weaning young animals at a time when their digestive tract is still immature presents a major challenge to ensuring the health and productivity of commercial herds [12,29,55]. Improving strategies to increase the efficiency of this transition would thus greatly benefit the swine industry.

## 4. Microbial Ecology and the Assembly of Gut Microbial Communities

### 4.1. Microbial Specialization and Cross-Feeding

As a result of limitations in the number of genes that can be encoded in a microbial genome [91], individual species tend to be very specialized. For heterotrophic microbial communities such as those of the gut environment, diversity tends to be high because a variety of different substrates is provided by the feed ingested by the animal host. Indeed, many microbial species may be involved at different steps in breaking down each type of substrate or, alternatively, different species may be competing for the same substrate [92]. As heterotrophic communities in the gut are anaerobic, the separation of metabolic tasks is very pronounced, as available terminal electron acceptors under these conditions have a lower reduction potential than oxygen, which results in lower efficiency in the extraction of energy from available substrates.

An important mechanism to ensure optimal efficiency in anaerobic environments is cross-feeding, as it allows for overcoming limitations from microbial specialization and reduced biochemical efficiency. For instance, the complete utilization of macromolecules from plant tissue or biomass, which represent the most commonly available substrates to gut symbionts, involves a multistep process that can only be accomplished through the combined metabolic activities of multiple individual members of a microbial community [93]. Since many metabolic reactions that take place under these conditions are thermodynamically more favorable when end products are maintained at low concentrations, these need to be eliminated from the system. Compounds that cannot be efficiently released into the environment or absorbed by the host have to be continuously metabolized by other members of the community in order to ensure efficiency of the system [57]. Cross-feeding, which can also be described in terms of trophic relationships [94], is an important core basic principle of microbial ecology, as it is based on dependencies amongst species with different metabolic activities. While cross-feeding relationships pertaining to carbohydrate utilization are generally of highest interest in gut microbiome research, dependencies related to other functions such as vitamin or nitrogen metabolism are likely of equal importance in shaping the species composition of a microbial community.

Compared to other anaerobic microbial habitats, the gut environment adds an additional level of complexity. Indeed, host cells can also provide substrates to microbial symbionts in the form of mucins or glycoproteins anchored at their surface, and they can also modulate the chemical conditions in the lumen by controlling the transport of ions and metabolites [95]. In turn, host cells respond to signals from the microbiota through feedback mechanisms. For instance, the expression of fucosylated glycans on the surface of intestinal epithelial cells that serve as attachment points for gut microbial species is stimulated by the presence of bacteria through activation of ERK and JNK signaling pathways in the host cells [96].

Together, substrates from ingested feed, host cell surface proteins, and secreted molecules, as well as microbial species and their end products, create localized micro-habitats with distinct combinations of chemical and biological parameters [97,98,99] (Figure 1). In turn, these sets of conditions act as selection factors that favor particular microbial species according to their encoded metabolic potential. This metabolic potential not only determines the ability to utilize available substrates, but also dictates the type of dependencies with other microbial species, such as the need to have end products metabolized or the need to acquire certain nutrients due to the absence of encoded synthesis pathway enzymes. The ability of a given species to thrive in a particular microbial habitat is then dependent on the degree to which its niche requirements are met by current micro-habitat conditions. As an example, niche selection is responsible for differences in microbial community composition amongst different compartments of the gut [100,101,102].

### 4.2. Resistance and Resilience Determine the Susceptibility of a Microbial Community to Dysbiosis

If factors such as substrate availability or the epithelial cell expression of surface molecules change, then a disruption of micro-habitat conditions may follow. In turn, these perturbations may affect the fit of microbial species to the micro-habitat, causing a change in composition favoring other microbial species better suited to thrive under the new conditions. Depending on its metabolic versatility and the extent of the disruptions, a microbial community may be able to resist a change in its composition [103,104]. However, if the perturbations go beyond the resistance capacity of the microbial community, then it may undergo a transition and enter a state of dysbiosis, where the representation of microbial species would be dramatically altered. If the original conditions are restored and the microbial community is sufficiently resilient, species composition may revert back to its previous state [103,104]. Alternatively, dysbiosis may persist, which can then further devolve to a diseased state.

Typically, gut microbial communities with high species diversity and high functional redundancy, such as what would be found in mature animals maintained under consistent diets and management practices [105], will tend to have high resistance and resilience. In contrast, gut microbial communities of young animals will tend to have lower resistance and resilience, as reduced species diversity and limited metabolic redundancies make them more susceptible to undergo profound changes in composition when challenged with abrupt transitions in diet formulation or disruptions such as stress [72,90]. Fluidity in gut species composition thus makes young animals more susceptible to dysbiosis and pathogen infection.

### 4.3. Colonization and Microbial Succession

Prior to birth, the gut of developing animals is mostly devoid of microbial life. As a result of exposure to microorganisms from the vaginal canal and from the external environment, gut microbiomes undergo rapid development as soon as animals are born. However, only microbial species with metabolic capabilities that are suited to the conditions of the neonate gut can colonize this environment and become established through proliferation. Indeed, of all the microbial species that the neonate is exposed to, only species with metabolic activities that match niche conditions of the gut at that time will be selected and become ‘pioneers’ [106].

However, the early colonized gut environment changes rapidly following the ingestion of colostrum and milk by the host, as well as from the accumulation of microbial end products generated by pioneer bacterial species. Combined with the activities of epithelial cells and their expression of surface molecules, the new sets of conditions trigger waves of microbial succession, i.e., successive peaks of proliferation by different groups of bacterial species that are each adapted to thrive in environments that were sequentially created during previous succession waves. Thus, microbiome development in young mammals is a continuous cycle of ‘niche preparation’, i.e., that the upcoming composition of microbial communities is determined by the current availability of dietary substrates, the composition of the current microbial community and its combined metabolic activities, as well as by epithelial cells and their modulation of their cell surface molecules and of the microbial environment. Current experimental data indicate that multiple events of microbial succession occur during early gut microbiome development [107].

At weaning, the abrupt diet change from milk, a liquid with easily digestible substrates, to solid feed, consisting primarily of plant-based ingredients that include polysaccharides, is likely the most disruptive event for the developing microbiome. In the context of the microbial ecology framework outlined above, this transition would result in a severe shift in micro-habitat conditions, with the metabolic capabilities of pre-weaning microbial populations not well suited to utilize polysaccharides and other plant-based substrates. Until stable bacterial consortia become established, microbial communities undergoing these transitions at weaning would be prone to dysbiosis, providing an opportunity for pathogens to proliferate and induce a diseased state. As weaning can also result in inflammation and stress, conditions that affect the physiology of host cells, host-induced factors can further exacerbate the risk of dysbiosis or prolong dysbiosis. In addition to the limited resistance of pre-wean microbial communities to the changes induced by weaning, there is also limited resilience as the conditions that selected for pre-wean microbial consortia would not be restored.

Another level of complexity during the post-weaning period would come from the development of more intricate networks of cross-feeding relationships as a result of the change in diet [88]. As the chemical composition of plant-based ingredients is very complex, a more diverse array of end products would be generated from substrate utilization, resulting in sequential cycles of changes in micro-habitat conditions, each consisting of the accumulation of different end products followed by the creation of new niche conditions that would then select for other sets of microbial symbionts to thrive. With every succession wave occurring during the post-weaning phase, the complexity of gut microbial communities would increase until a mature state is reached [108,109]. Higher species diversity and functional redundancy would bring higher resistance and resilience [107].

## 5. Factors That Impact Gut Microbial Community Assembly and Composition

### 5.1. Diet

As the main source of substrates for the growth of gut microorganisms, diet is the most impactful factor that shapes the composition and hence the function of gut microbiomes [110,111,112] (Figure 2). Indeed, early gut microbiota development in pigs can be divided into three main phases according to diet: neonatal, nursing, and post weaning. The neonatal phase would consist of the relatively brief period prior to nursing, during which dietary substrates have yet to be provided to the newborn piglet; the only available source of nutrients in the gut for colonizing bacterial species would be mucus and other secretions, as well as molecules expressed on the surface of epithelial cells. During the nursing phase, colostrum and milk become the first sources of dietary substrates for the developing gut microbiome [113]. Since the main function of milk is to provide easily digestible nutrients to the neonate, the range of available substrates for symbiotic microorganisms is relatively limited, with the main source consisting primarily of milk glycans. With weaning, there is an abrupt transition from a narrow set of animal glycans to a broad array of plant glycans of high architectural diversity, as well as to an abundance of more recalcitrant substrates such as structural polysaccharides.

### 5.2. Environment

Starting at birth, young animals are in constant exposure to a wide variety of microorganisms that range from the various microbiomes of the sow (e.g., vaginal, skin, and fecal, as well as colostrum and milk) to the surrounding physical environment (Figure 2). While a consensus on the exact contribution of these different pools of microorganisms during colonization and microbial succession has yet to be clearly defined, their role as a source of gut microbial symbionts has been well established [80,114,115,116,117,118,119,120]. Control of microbial populations in these sources could potentially become part of future strategies to modulate early gut microbiome development.

### 5.3. Host

Various factors associated with the host, such as behavior and physiology, can impact development of the gut microbiome [121]. For instance, young pigs commonly experience a dramatic reduction in feed intake at weaning, which can be caused by poor adaptation to solid feed or to stress induced from separation anxiety, establishment of a new hierarchical order, or adapting to unfamiliar surroundings. Regardless of the cause, reduced feed intake can induce intestinal inflammation [7,14], which triggers the production of reactive oxygen and nitrogen species by host cells [122]. One of these compounds, nitric oxide (NO), can be transformed into nitrate (NO_3_^−^) when it is released in the intestinal lumen, providing an advantage to bacterial species that can express nitrate reductase [123,124]. Notably, pathogens such as *Salmonella enterica serovar Typhimurium* or enterotoxigenic *Escherichia coli* (ETEC) take advantage of this metabolic activity by inducing inflammation, which not only provides them with nitrates to promote their growth, but also facilitates the transition to a state of dysbiosis [125,126,127].

Genetics can also have an effect on microbiome development and composition. There is strong evidence for the evolutionary conservation of mechanisms that determine early gut microbiome development amongst mammals, particularly during the early stages of bacterial colonization [107,128,129,130,131,132]. In addition, while less impactful than diet, breed was reported to have an effect on gut bacterial composition, as was observed either through direct comparison of purebred pig lines [133,134] or using a cross-fostering system [135]. While the mechanisms responsible for these effects remain to be further elucidated, differences in glycosylation patterns amongst breeds [136,137] could provide a possible explanation for the modulation of gut microbial community composition by host cells. Glycosylation is a post-translational process that covalently attaches oligosaccharides to either asparagine or serine/threonine, to create N-linked or O-linked glycans, respectively [138]. A number of gut microbial species have the ability to metabolize oligosaccharides from host N- and O-glycans, such as those found in milk proteins and mucins, respectively [139] (Figure 2). In light of their complex architecture, polymorphic nature, and differential expression patterns, glycans and their oligosaccharides exhibit characteristics of factors with the potential to modulate distinct gut microbiome composition amongst breeds or individuals [88]. Notably, mucins are particularly attractive candidates for this function, with at least 20 different encoding genes identified [140,141] that each contain eight possible O-glycan cores that are distributed amongst a variable number of tandem repeats [142]. The regulation of mucin glycosylation is itself modulated in part by gut microbial species, as the presence of particular symbionts is required for the expression of specific glycosyltransferases [143].

## 6. Colonization and Succession Events of the Developing Swine Gut Microbiome

### 6.1. Early Colonization and Microbial Succession

While it has been reported that microbial colonization of the gut could be initiated as early as in utero [144], the validity of this mechanism has since been disputed [145,146]. Thus, the first dominant bacterial group to become established in the swine gut after birth consists of facultative anaerobes affiliated with Enterobacteriaceae, such as members of the *Escherichia*/*Shigella* genera [107,108]. This is very similar to what has been described in humans, where this group is responsible for metabolizing oxygen, thereby creating a suitable anaerobic niche for the establishment of the subsequent prominent groups of gut symbionts [147,148,149,150,151,152].

The second dominant microbial group to emerge in the swine gut includes members of the genus *Bacteroides* [108], whose main signature metabolic capability is the utilization of complex oligosaccharides from milk [107,153]. Gut *Bacteroides* establish cross-feeding relationships with other SCFA-producing species such as *Faecalibacterium prausnitzii* [154,155], *Anaerostipes caccae* [156], and *Eubacterium ramulus* [157], as well as with the *Subdoligranulum variabile-Hungatella hathewayi* consortium [158]. Members of *Clostridium sensu stricto* also become prominent during this early microbiome development period [107,110,159]. Consistent with milk providing nutrients for the development of the gut microbiome, gut metagenomic analyses from nursing pig samples have revealed a high representation of proteins and enzymes needed for metabolizing glycans from milk, such as sialidase (EC 3.2.1.18) and beta-hexosaminidase (EC 3.2.1.52) [110], as well as gene families involved in the uptake and utilization of lactose and galactose [160]. It is likely that the reported consistency in microbial composition during these early stages is at least in part due to the common sources of substrates available for microbial growth in the gut, as they originate from the neonate and the sow, with genetics perhaps more likely to have a greater impact than environmental effects or management practices.

### 6.2. Post-Weaning Microbial Succession

As a result of the abrupt changes that take place during weaning, the composition of gut microbial communities is dramatically altered during this period [108,109]. More diverse patterns of gut microbial compositions have been reported during the post-weaning period compared to earlier stages of development, which has made the identification of common or core bacterial groups during weaning very challenging. This increased variation can likely be attributed at least in part to the wide range of dietary strategies that can be implemented to help the gut of weaned pigs in adapting to new diets [161]. For instance, the inclusion of specialty ingredients such as fishmeal, whey, or oats likely has a significant impact on the types of bacterial species that become favored in the gut environment [86].

Thus, in contrast to the predominant microbial species of the nursing phase, bacterial groups that can metabolize plant polysaccharides and plant glycans, such as Prevotellaceae [162], increase in abundance in response to the change in diet. Accordingly, coding sequences for enzymes involved in the breakdown of plant-derived polymers, such as endo-1,4-β-xylanases (EC 3.2.1.8), α-N-arabinofuranosidases (EC 3.2.1.55), mannase, as well as β-xylosidases (EC 3.2.1.37), have been reported in higher abundance in the gut of weaned pigs [109,110,160,163]. There is also an increase in the levels of other microbial groups that can metabolize smaller compounds that are either present in the feed (e.g., plant monosaccharides such as tagatose) or produced as end products from other microbial symbionts (e.g., lactate) into SCFAs. These specialists include members of families such as Veillonellaceae (e.g., *Megasphaera*) or Oscillospiraceae (e.g., *Faecalibacterium*, *Subdoligranulum*). Other bacterial groups also reported as predominant during the post-weaning period have included Ruminococcaceae, Lachnospiraceae, and Lactobacillaceae [72,110,160]. As species affiliated with *Lactobacillus* are known to metabolize starch as a substrate, peaks of abundance during the early post-weaning period may be due to the delayed expression of host alpha-amylase, allowing the availability of dietary starch to gut microbial species in newly weaned pigs [164]. The early post-weaning period is also prone to increased incidences of diarrhea, reflecting the higher risk of dysbiosis as a result of the transition to a new diet and increased stress; diarrhea has been associated with a higher relative abundance of species affiliated with taxa such as *Sutterella*, *Campylobacter,* and Fusobacteriaceae [165].

## 7. Weaning as a Transition Period for the Establishment of a Mature Microbiome

From the perspective of microbiome development, establishing bacterial species that will become the prominent members of mature and stable gut microbial communities should begin as early as possible during the post-weaning period. As these species need to be integrated during the assembly of microbial communities, understanding their metabolic capabilities and functions is thus critical to developing strategies that promote their establishment and proliferation during the post-weaning period. Earlier establishment of a mature microbiota would also be beneficial by reducing the number of microbial succession events, thus building the resistance and resilience of the developing microbiota. Considering the evidence of crosstalk or cross-regulation between microbiota and epithelial cells [96,143,166], earlier establishment of a mature microbiota could also facilitate maturation of the gut and immune system.

In light of the complexities of mature gut microbiomes, one of the current challenges in the field is the identification of these important microbial species, many of which may be unknown or yet to be characterized [88,89,167]. Recently, our group has reported on the fecal bacterial communities of finishing barrows [168] and determined that seven Operational Taxonomic Units (OTUs) represented, on average, 42.8% of sequence reads in the samples analyzed (Table 1). Considering their predominance in finishing pigs, it would be reasonable to hypothesize that the bacterial species corresponding to these OTUs would be important members of mature swine gut bacterial communities. With the exception of one OTU that was not detected in one nursery study, all seven OTUs that were predominant in finishing barrows had also been found in nursery pigs [169,170,171], but in much lower abundance (Table 1). Notably, five of these OTUs could not be reliably assigned to known bacterial species based on 16S rRNA sequence identity, indicating that they likely correspond to bacterial species that have yet to be cultured or characterized (Table 2). Considering that 16S rRNA gene sequences from public databases were found to be a very close match to these OTUs (Table 2), their corresponding unknown species have been previously found by other research groups [172,173,174,175,176], indicating that they may be common residents of the swine gut across a range of different geographical areas. Gaining further insight on the metabolic functions or potential of the bacterial species corresponding to these OTUs, as well as others that are prominent in mature pigs, could allow the development of strategies to accelerate maturation of the swine gut microbiome at the weaning stage.

## 8. Concluding Remarks and Future Outlook

Early weaning presents major challenges to young pigs, as they need to adapt to a completely different diet regimen, as well as adjust to new environmental conditions and social structures, while being equipped with only an immature immune system and under-developed intestinal organs. The abrupt change in diet also completely disrupts the composition of the gut microbiome in post-weaned pigs, resulting in an increased susceptibility to dysbiosis, thus predisposing young animals to gut dysfunction and disease. Together, these challenges can not only compromise health and performance during the nursery phase, but also impact their progression later during the productive stages of their life [32,38,177].

Considering that individual gut bacterial species are specialists that can be integrated as part of microbial communities when they are provided with conditions that meet their niche requirements, it is critical to gain more insights on the metabolic capabilities of beneficial microbial species if we aim to accelerate their establishment and integration as part of mature gut microbiota [178]. Achieving a mature state by minimizing the number of microbial succession events during the post-weaning period would reduce the risk of dysbiosis by increasing the resistance and resilience of the gut microbiome. Accelerating the development of a mature or stable gut microbiome during the post-weaning period could also provide other benefits, such as earlier maturation of the immune system and digestive tract.

## Figures and Tables

**Figure 1 microorganisms-11-01753-f001:**
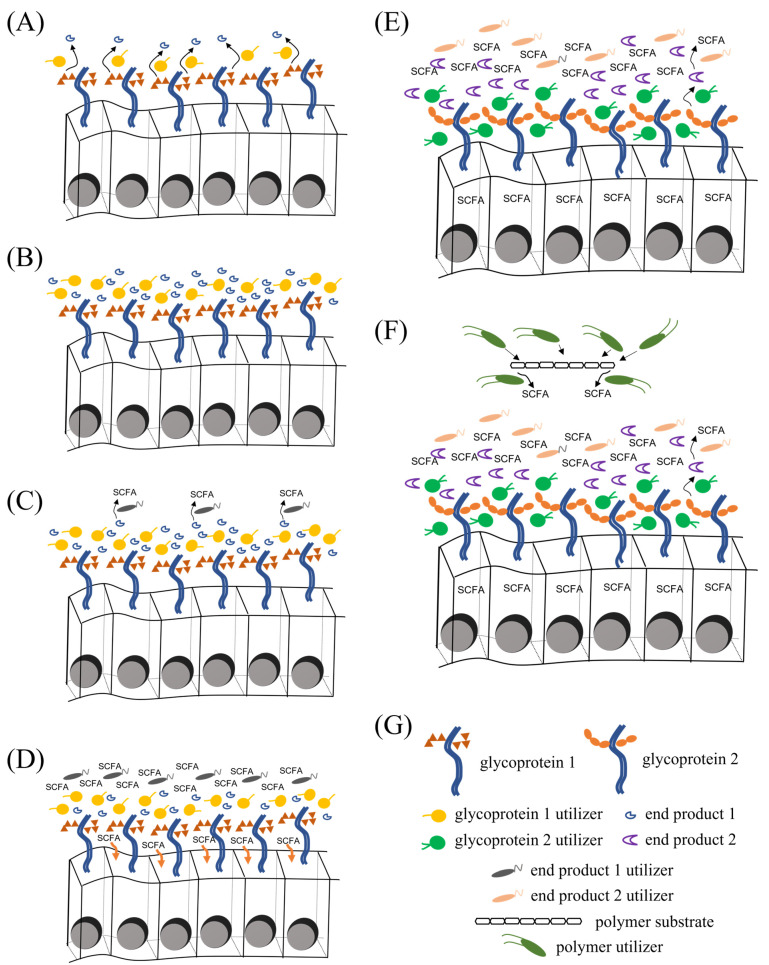
Hypothetical model of micro-habitat formation and microbial succession. (**A**) The surface protein ‘glycoprotein 1’, which is expressed by epithelial cells, can be metabolized by a specialized microbial species (‘glycoprotein utilizer 1’) to generate a particular compound (‘end product 1’). (**B**) As ‘glycoprotein utilizer 1’ proliferates, there is an accumulation of ‘end product 1’. (**C**) The compound ‘end product 1’ is in turn metabolized into short-chain fatty acids (SCFAs) by a specialized cross-feeding species (‘end product 1 utilizer’). (**D**) As ‘end product 1 utilizer’ proliferates, there is an accumulation of SCFAs, which are absorbed by epithelial cells. (**E**) As a result of SCFA absorption, epithelial cells switch from expressing ‘glycoprotein 1’ to expressing ‘glycoprotein 2’; this change in glycoprotein expression results in a complete transformation of the bacterial community composition, from primary utilizers to cross-feeding species. (**F**) An additional layer of complexity comes from substrates provided by the diet (‘polymer substrate’), which result in the proliferation of other microbial specialists that each can metabolize a particular type of dietary substrate. These utilizers of dietary substrates can also support other consortia of cross-feeding species (not represented in this diagram). (**G**) Figure legend.

**Figure 2 microorganisms-11-01753-f002:**
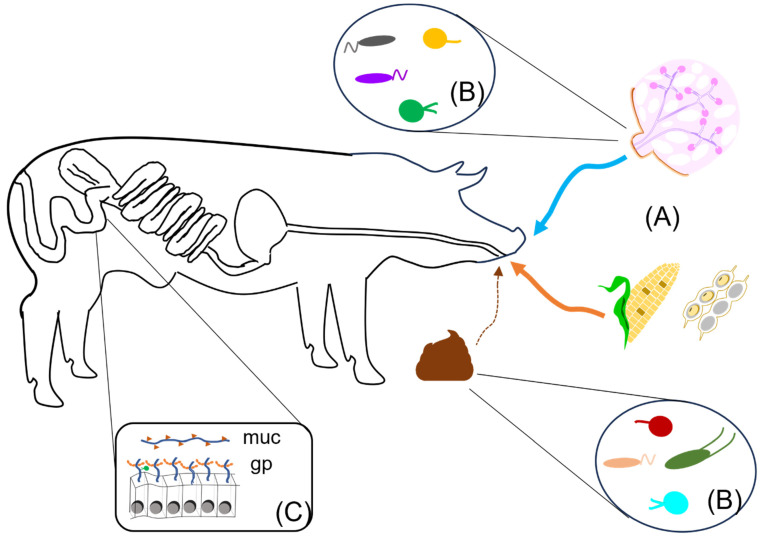
Factors that impact gut microbial community assembly and composition. Diet is the most influential factor affecting the assembly and composition of gut microbial communities in young pigs (**A**); milk (nursing phase—blue arrow) and plant-based ingredients, such as corn and soybean meal (post-weaning phase—orange arrow), provide very different types of substrates that each support distinct types of metabolic activities. The young pig’s environment contains different microbiomes (**B**), such as those found on the sow’s teats, milk, and feces, which act as a source of microorganisms that can colonize the gut. The developing gastrointestinal tract also plays an important role, such as (**C**) through the expression of host factors, which include mucins (muc) and cell-bound glycoproteins (gp).

**Table 1 microorganisms-11-01753-t001:** Comparison of relative abundance (%) at two different production stages of the most prominent OTUs identified in finishing pigs.

OTUs	Finishing ^a^	Nursery 1 ^b^	Nursery 2 ^c^	Nursery 3 ^d^
Ssd-0675	1.98	0.0040	0.49	0.16
Ssd-1048	3.08	0.0001	0.03	0.05
Ssd-1079	1.54	0.0037	0.98	0.69
Ssd-1085	4.98	0.0119	0.14	0.38
Ssd-1095	24.23	0.0057	0.11	0.49
Ssd-1115	4.18	0.0008	0.03	0.25
Ssd-1144	2.89	0	0.02	0.04

For finishing pigs, means were calculated from all barrows (light and heavy). (a) Fowler et al. (2023) [168]. For nursery pigs, means were calculated only from samples collected from controls. (b) Poudel et al. (2020) [169]. (c) Fresno Rueda et al. (2021) [171]. (d) Poudel et al. (2022) [170].

**Table 2 microorganisms-11-01753-t002:** 16S rRNA sequences from uncultured bacteria that were a close match ^1^ to prominent OTUs identified in finishing pigs.

OTUs	Accessions (Genbank)	Closest Relative (Id%) ^2^
Ssd-0675	KM365293.1 ^a^, AF371834.1 ^b^	*Christensenella massiliensis* (84.5%)
Ssd-1048	HQ716448.1 ^c^, GU619382.1 ^d^, KF520973.1 ^e^	*Caecibacteroides pullorum* (86.9%)
Ssd-1079	HQ716578.1 ^c^, KF518096.1 ^e^	*Mahella australiensi* (83.0%)
Ssd-1095	AF371926.1 ^b^, HQ716393.1 ^c^, AB506368.1 ^f^	*Lignipirellula cremea* (80.9%)
Ssd-1115	GU605560.1 ^d^	*Treponema peruense* (84.6%)

1. 16S rRNA sequences from the NCBI ‘nt’ database showing at least 99% sequence identity to their corresponding OTU. 2. Nucleotide sequence identity (%) between each OTU and its corresponding closest valid relative. (a) Unpublished data; (b) Leser et al. (2002) [172]; (c) Kalmokoff et al. (2011) [173]; (d) Jeong et al. (2011) [174]; (e) Li et al. (2014) [175]; (f) Kobayashi et al. (2011) [176].

## Data Availability

No new data were created or analyzed in this study. Data sharing is not applicable to this article.

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
