# Peer review of "Impact of Early Weaning on Development of the Swine Gut Microbiome"

_microorganisms, 2023, doi:10.3390/microorganisms11071753_

Round 1
Reviewer 1 Report
This manuscript reviewed the effects of early weaning on intestinal development and intestinal microbiome of pigs, which has important guiding significance for pig production. This paper consists of two parts, the first is the effect of early weaning on the intestinal development of pigs, and the second is the effect of early weaning on the intestinal flora of pigs. However, the proportion of the two parts is seriously unbalanced, and most of the paper is devoted to the effect of early weaning on the intestinal flora of pigs. Therefore, I suggest that the author adjust the structure of the paper to make the proportion of the two parts relatively balanced.
none.
Author Response
R1-C1. This manuscript reviewed the effects of early weaning on intestinal development and intestinal microbiome of pigs, which has important guiding significance for pig production. This paper consists of two parts, the first is the effect of early weaning on the intestinal development of pigs, and the second is the effect of early weaning on the intestinal flora of pigs. However, the proportion of the two parts is seriously unbalanced, and most of the paper is devoted to the effect of early weaning on the intestinal flora of pigs. Therefore, I suggest that the author adjust the structure of the paper to make the proportion of the two parts relatively balanced.
Author reply: After reading the reviewer’s comment, we see how the title gives the impression that the content of the review is not well balanced. As a solution, we have revised the title to:
“Impact of early weaning on development of the swine gut microbiome”.
Reviewer 2 Report
Please provide bibliography - Microbiome research has been revolution- 207 ized by the rapid development of different ‘omics’-type tools and platforms, as they have 208 allowed more comprehensive investigations of complex microbial communities. While 209 great insights have been achieved so far, further investigations are still required to gain a 210 deeper understanding of microbiome capabilities towards developing strategies to mod- 211 ulate their function for optimizing livestock production and animal health.-
Please provide bibliography -In light of the increased recognition that early events taking place dur- 215 ing microbiome development can affect its function at later stages, gaining a better under- 216 standing of the mechanisms that dictate or affect development of gut microbiomes in 217 young animals has been of very high interest. This is particularly true in food animal pro- 218 duction systems, where ensuring health of young animals while conforming to stricter 219 restrictions on traditional management practices such as antibiotics use remains an ongo- 220 ing challenge. Within the swine industry, the standard practice of abruptly weaning 221 young animals at a time when their digestive tract is still immature presents a major chal- 222 lenge to ensuring the health and productivity of commercial herds. Improving strategies 223 to increase the efficiency of this transition would thus greatly benefit the swine industry
Please provide bibliography -Typically, gut microbial communities with high species diversity and high functional 289 redundancy, such as would be found in mature animals maintained under consistent diets 290 and management practices, will tend to have high resistance and resilience. In contrast, 291 gut microbial communities of young animals will tend to have lower resistance and resil- 292 ience, as reduced species diversity and limited metabolic redundancies make them more 293 susceptible to undergo profound changes in composition when challenged with abrupt 294 transitions in diet formulation or disruptions such as stress. Fluidity in gut species com- 295 position thus makes young animals more susceptible to dysbiosis and pathogen infection.
Please insert more recent published articles.
Please also specify that this is a descriptive review.
Moderate english revision.
Author Response
Reviewer 2
R2-C1. Please provide bibliography “Microbiome research has been revolutionized by the rapid development of different ‘omics’-type tools and platforms, as they have allowed more comprehensive investigations of complex microbial communities. While great insights have been achieved so far, further investigations are still required to gain a deeper understanding of microbiome capabilities towards developing strategies to modulate their function for optimizing livestock production and animal health.”
Author reply: References were added as recommended (highlighted in the revised manuscript for your convenience, also marked with comment box):
“Microbiome research has been revolutionized by the rapid development of different ‘omics’-type tools and platforms, as they have allowed more comprehensive investigations of complex microbial communities [88,89]. While great insights have been achieved so far, further investigations are still required to gain a deeper understanding of microbiome capabilities towards developing strategies to modulate their function for optimizing livestock production and animal health [63].”
- Flint, H.J.; Scott, K.P.; Louis, P.; Duncan, S.H. The role of the gut microbiota in nutrition and health. Nat Rev Gastroenterol Hepatol 2012, 9, 577–589.
- Gaio, D.; DeMaere, M.Z.; Anantanawat, K.; Chapman, T.A.; Djordjevic, S.P.; Darling, A.E. Post-weaning shifts in microbiome composition and metabolism revealed by over 25 000 pig gut metagenome-assembled genomes. Microb Genom 2021, 7, 000501.
- Dong B; Lin X; Jing X; Hu T; Zhou J; Chen J; Xiao L; Wang B; Chen Z; Liu J; et al. A bacterial genome and culture collection of gut microbial in weanling piglet. Microbiol Spectr 2022, 10, e0241721.
R2-C2. Please provide bibliography “In light of the increased recognition that early events taking place during microbiome development can affect its function at later stages, gaining a better understanding of the mechanisms that dictate or affect development of gut microbiomes in young animals has been of very high interest. This is particularly true in food animal production systems, where ensuring health of young animals while conforming to stricter restrictions on traditional management practices such as antibiotics use remains an ongoing challenge. Within the swine industry, the standard practice of abruptly weaning young animals at a time when their digestive tract is still immature presents a major challenge to ensuring the health and productivity of commercial herds. Improving strategies to increase the efficiency of this transition would thus greatly benefit the swine industry.”
Author reply: References were added as recommended (highlighted in the revised manuscript for your convenience, also marked with comment box):
“In light of the increased recognition that early events taking place during microbiome development can affect its function at later stages [63, 67], gaining a better understanding of the mechanisms that dictate or affect development of gut microbiomes in young animals has been of very high interest. This is particularly true in food animal production systems, where ensuring health of young animals while conforming to stricter restrictions on traditional management practices such as antibiotics use remains an ongoing challenge [55]. Within the swine industry, the standard practice of abruptly weaning young animals at a time when their digestive tract is still immature presents a major challenge to ensuring the health and productivity of commercial herds [12, 29, 55]. Improving strategies to increase the efficiency of this transition would thus greatly benefit the swine industry.”
- Jayaraman, B.; Nyachoti, C.M. Husbandry practices and gut health outcomes in weaned piglets: A review. Anim Nutr 2017, 3, 205-211.
- Pluske, J.R. Invited review: Aspects of gastrointestinal tract growth and maturation in the pre- and postweaning period of pigs. J Ani Sci 2016, 94, 399-411.
- Zheng, L.; Duarte, M.E.; Sevarolli Loftus, A.; Kim, S.W. Intestinal health of pigs upon weaning: Challenges and nutritional intervention. Front Vet Sci 2021, 8, 628258.
- Flint, H.J.; Scott, K.P.; Louis, P.; Duncan, S.H. The role of the gut microbiota in nutrition and health. Nat Rev Gastroenterol Hepatol 2012, 9, 577–589.
- Tamburini, S.; Shen, N.; Wu, H.C.; Clemente, J.C. The microbiome in early life: Implications for health outcomes. Nat Med 2016, 22, 713-722.
R2-C3. Please provide bibliography “Typically, gut microbial communities with high species diversity and high functional redundancy, such as would be found in mature animals maintained under consistent diets and management practices, will tend to have high resistance and resilience. In contrast, gut microbial communities of young animals will tend to have lower resistance and resilience, as reduced species diversity and limited metabolic redundancies make them more susceptible to undergo profound changes in composition when challenged with abrupt transitions in diet formulation or disruptions such as stress. Fluidity in gut species composition thus makes young animals more susceptible to dysbiosis and pathogen infection.”
Author reply: References were added as recommended (highlighted in the revised manuscript for your convenience, also marked with comment box):
“Typically, gut microbial communities with high species diversity and high functional redundancy, such as would be found in mature animals maintained under consistent diets and management practices [105], will tend to have high resistance and resilience. In contrast, gut microbial communities of young animals will tend to have lower resistance and resilience, as reduced species diversity and limited metabolic redundancies make them more susceptible to undergo profound changes in composition when challenged with abrupt transitions in diet formulation or disruptions such as stress [72,90]. Fluidity in gut species composition thus makes young animals more susceptible to dysbiosis and pathogen infection.”
- Dou, S.; Gadonna-Widehem, P.; Rome, V.; Hamoudi, D.; Rhazi, L.; Lakhal, L.; Larcher, T.; Bahi-Jaber, N.; Pinon-Quintana, A.; Guyonvarch, A.; et al. Characterisation of early-life fecal microbiota in susceptible and healthy pigs to post-weaning diarrhoea. PLOS One 2017, 12, e0169851.
- Guevarra, R.B.; Lee, J.H.; Lee, S.H.; Seok, M.-J.; Kim, D.W.; Kang, B.N.; Johnson, T.J.; Isaacson, R.E.; Kim, H.B. Piglet gut microbial shifts early in life: causes and effects. J Animal Sci Biotechnol 2019, 10, 1.
- Qi, K.; Men, X.; Wu, J.; Deng, B.; Xu, Z. Effects of growth stage and rearing pattern on pig gut microbiota. Curr Microbiol 2022, 79, 136.
R2-C4. Please insert more recent published articles.
Author reply: Agreed. In the revised manuscript, we added:
- 2 articles published in 2023 (references 65, 140]
- 11 articles published in 2022 [references 64, 85, 89, 105,108, 109, 121, 137, 162, 168, 179]
- 3 articles published in 2021 [references 77, 84, 88]
- 1 article published in 2020 [reference 134].
Please see the revised manuscript – the added references are highlighted in blue in the reference list as well as when cited in the text.
R2-C5. Please also specify that this is a descriptive review
Author reply: We added this statement to the abstract of the revised manuscript:
“(…), and this descriptive review aims to present a microbial ecology-based perspective on these events.”
Reviewer 3 Report
Benoit St-Pierre et al. summarized the present microbiome studies about the role of early weaning in pigs. Early weaning in pigs typically done between two to four weeks of age, presents challenges due to the sudden separation from the sow, dietary changes, and social hierarchy adjustments. The immature gut during this transition undergoes regression in structure and function, affecting digestion and barrier integrity. The gut microbiota experiences significant alterations during this period, leading to instability and potential dysbiosis. This review proposed that promoting gut microbiome maturation, identifying key microbial species and facilitating their establishment in early post-weaning communities could be a potential strategy. It is an important and timely review, and I only have some minor comments.
Line 310. Change ‘short chain fatty acids (SCFA)’ to short chain fatty acids (SCFAs)
Line 357. It would be better to use a diagram to show ‘Factors that impact gut microbial community assembly and composition’.
Author Response
Reviewer 3
Benoit St-Pierre et al. summarized the present microbiome studies about the role of early weaning in pigs. Early weaning in pigs typically done between two to four weeks of age, presents challenges due to the sudden separation from the sow, dietary changes, and social hierarchy adjustments. The immature gut during this transition undergoes regression in structure and function, affecting digestion and barrier integrity. The gut microbiota experiences significant alterations during this period, leading to instability and potential dysbiosis. This review proposed that promoting gut microbiome maturation, identifying key microbial species and facilitating their establishment in early post-weaning communities could be a potential strategy. It is an important and timely review, and I only have some minor comments.
R3-C1. Line 310. Change ‘short chain fatty acids (SCFA)’ to short chain fatty acids (SCFAs)
Author reply: Change made – please see revised manuscript. We also added “SCFA” in the main text (line 192, revised manuscript), because we realized that the abbreviation is used later in the main text, but it was not defined in its first instance in the previous version of the manuscript.
R3-C2. Line 357. It would be better to use a diagram to show ‘Factors that impact gut microbial community assembly and composition’.
Author reply: Thank you for the recommendation; the diagram (Figure 2) is a great addition to the manuscript. Please see the revised manuscript for the figure.
Figure 2. Factors that impact gut microbial community assembly and composition. Diet is the most influential factor affecting the assembly and composition of gut microbial communities in young pigs (A); milk (nursing phase – blue arrow) and plant-based ingredients, such as corn and soybean meal (postweaning phase – orange arrow), provide very different types of substrates that each support distinct types of metabolic activities. The young pig’s environment contains different microbiomes (B), such as found on the sow’s teats, milk and feces, which act as a source of microorganisms that can colonize the gut. The developing gastrointestinal tract also plays a role, such as (C) through the expression of host factors, which include mucins (muc) and cell-bound glycoproteins (gp).